# *Cassia alata* L.: A Study of Antifungal Activity against *Malassezia furfur*, Identification of Major Compounds, and Molecular Docking to Lanosterol 14-Alpha Demethylase

**DOI:** 10.3390/ph17030380

**Published:** 2024-03-16

**Authors:** Nyi Mekar Saptarini, Resmi Mustarichie, Silviana Hasanuddin, Mary Jho-Anne Tolentino Corpuz

**Affiliations:** 1Department of Pharmaceutical Analysis and Medicinal Chemistry, Faculty of Pharmacy, Universitas Padjadjaran, Sumedang 45363, West Java, Indonesia; resmi.mustarichie@unpad.ac.id; 2Department of Pharmacy, Universitas Mandala Waluya, Kendari 93561, Southeast Sulawesi, Indonesia; silviana.hasanuddin@gmail.com; 3Department of Pharmacy, Faculty of Pharmacy, University of Santo Tomas, Manila 1015, Philippines; mtcorpuz@ust.edu.ph

**Keywords:** candle bush, stearidonic acid, concentration-dependent, homology modeling

## Abstract

Empirically, in Indonesia, the leaves of *Cassia alata* L. (candle bush or ketepeng cina) have been used as a topical antifungal agent. *Malassezia furfur* is a natural microorganism found in the human body. It is among the factors contributing to conditions such as pityriasis versicolor, a common, benign, superficial fungal infection of the skin that is closely associated with seborrheic dermatitis and dandruff. This study aimed to explore *C. alata* leaves, starting from determining antifungal activity against *M. furfur* and the identification of major compounds in the ethyl acetate and n-hexane fractions, and then we carried out molecular docking of the major compounds in the n-hexane fraction to lanosterol 14-alpha demethylase. The method was the disc diffusion technique to test antifungal activity, LC-MS/MS for major compound identification, and homology modeling through Swiss Models for molecular docking. The fractions of ethyl acetate and n-hexane extract showed concentration-dependent antifungal activity against *M. furfur*. The LCMS/MS analysis revealed five major compounds in the ethyl acetate and n-hexane fractions. The molecular docking demonstrated the highest binding affinity with stearidonic acid at −7.2 kcal/mol. It can be concluded that the compounds in the n-hexane fraction have antifungal activity against *M. furfur*, as supported by both in vitro and in silico studies.

## 1. Introduction

Medicinal plants are becoming popular due to their availability in local communities, natural origins, cheaper purchase, and ease of administration. Herbal plants may be a useful alternative treatment if there are many side effects and drug resistance [1,2]. Indonesian communities use *Cassia alata* L., also known as *Senna alata* L. or candle bush (English) or ketepeng Cina (Indonesian) as a traditional remedy for diverse skin conditions [3]. The World Health Organization (WHO) has mentioned that 80% of the population in developing countries is dependent on herbal medicines [4]. About one-third of used medicines are obtained from natural sources. This has resulted in the documentation of approximately 40,000–70,000 species of medicinal plants with therapeutic potential [5]. *C. alata* is taxonomically classified into Plantae (kingdom), Fabales (order), Fabaceae (family), Caesalpinioideae (subfamily), Cassia (genus), and *C. alata* (species). This plant is a flowering shrub, called “candle bush” because of its flower framework, with a height of about 1 to 4 m, growing in sunlit and humid areas. The leaves are oblong with five to fourteen leaflet sets, robust petioles (2 to 3 mm), and dense flowers (20 × 50 by 3 × 4 cm). The zygomorphic flowers are a bright yellow color with seven stamens and a pubertal ovary. The fruit is a tetragonal pod measuring 10 to 16 × 1.5 cm, with thick, flattened wings, and brown when ripe, with many diamond-shaped brown seeds [6]. The *C. alata* plant can be seen in Figure 1 [7]. In Southeast Sulawesi, this plant is traditionally applied to alleviate fungal-induced skin itch by grinding and topically applying it to the affected area [3]. The aqueous extract of *C. alata* has antifungal activity against several fungal isolates, including dermatophytes (*Trichophyton mentagrophytes* and *Microsporum canis*) [8,9]. Hassan et al. [8] stated that the active compound of *C. alata* leaf extract inhibits fungal cell walls due to the formation of pores in cells and leakage of cytoplasmic constituents. A study by Edegbo et al. [9] showed that n-hexane extract has higher antifungal activity than aqueous, chloroform, methanol, and ethanol extracts and is most active against *Epidermophyton floccosum* and *M. canis*, while Sujatha et al. [10] reported that the n-hexane extract of *C. alata* leaves has antifungal activity against *M. canis, M. audouinii, T. mentagrophytes, T. rubrum*, and *E. floccosum* at various concentrations. The chloroform extract of *C. alata* leaves was also reported to have antifungal activity against fungal isolates, including *E. floccosum, M. gypseum*, and *T. mentagrophyte* [9,11]. Meanwhile, Sule et al. [12] stated that the ethanol extract of *C. alata* leaves has antifungal activity against *M. canis, T. mentagrophytes, T. verrucosum,* and *E. fluccossum*. The methanol extract of *C. alata* leaves was also proven to have antifungal activity against *M. gypseum* and *T. rubrum* [13]. *C. alata* leaves contain steroids, alkaloids, saponins, flavonoids, terpenoids [14], and anthraquinones [15]. *C. alata* is known to harbor an array of bioactive compounds within its leaves. Flavonoids in leaves that have been identified include flavonols and flavones [16]. Prior structure identifications conducted through IR and NMR analyses have indicated the presence of flavonol-type compounds, specifically kaempferol and kaempferol-3-O-β-D-glucopyranoside [17]. In the leaves of *C. alata,* anthraquinones (alatinone and alatonal), phenolic compounds (rhein, chrysphanol, kaempferol, aloeemodin, and glycosides), fatty acids (linoleic, oleic, and palmitic acid), steroids, and terpenoids (sitosterol, stigmasterol, and campesterol) have been identified [18,19].

Seborrheic dermatitis is a skin disorder that commonly manifests in body regions abundant in sebaceous glands, including the chest, face, and scalp [20]. Malassezia is widely believed to induce a nonspecific immune response that may lead to alterations in skin functionality among individuals grappling with this condition [21]. *Malassezia furfur* is a member of a monophyletic genus of fungi found on the human skin. This fungus usually accounts for more than 80% of the total fungal population on human skin and is often isolated in both healthy and sick hosts [22]. The overall occurrence of seborrheic dermatitis ranges from 1 to 3% in the general population and reaches as high as 34 to 83% among those with compromised immune systems [23]. The risk factors associated with this disease include age, gender, increased sebaceous gland activity, immune deficiency, neurological and psychiatric disorders, the utilization of specific medications, and low environmental humidity or temperature [24]. These alterations elicit an inflammatory response within the host. Several commonly used treatments often fail to definitively stop seborrheic dermatitis, thereby increasing the likelihood of its recurrence. External factors such as weather fluctuations and stress also exert influence [25].

Treatments for seborrheic dermatitis, including keratolytic agents, aid in the removal of the hyperproliferative outer layer of the stratum corneum [26]. Antifungal agents target Malassezia, while anti-inflammatories such as corticosteroids effectively mitigate the skin’s inflammatory response [27]. Ketoconazole, an imidazole derivative from the azole group of antifungals, inhibits CYP51A1 by forming a coordinate bond between the nucleophilic nitrogen of the azole heterocycle (N3 imidazole and N4 triazole) and the heme iron in the iron state of the enzyme active site. Inhibition of lanosterol 14-alpha demethylase causes accumulation of 14-alpha-methyl sterol on the surface of the fungus and changes in the permeability and stiffness of the plasma membrane, resulting in the inhibition of fungal growth. The selective inhibition of fungal CYP51A1 is essential to improve the therapeutic index [28].

Concerns center around drug resistance, noncompliance, and potential side effects; for example, prolonged corticosteroid use can lead to adverse effects and compromise patient adherence [29]. Similarly, antifungal use can give rise to side effects such as a burning sensation, skin redness, and hair loss or alopecia [30]. As a result, many studies have shifted toward exploring natural active ingredients sourced from plants, animals, and minerals. This study focused on using plants as active ingredients in pharmaceutical development, which has great potential to produce new compounds with different structures and bioactivity. Several recent studies have explored the efficacy of *C. alata* as a prospective therapeutic agent for treating skin infections. This study is important to prove the antifungal activity of *C. alata* which is empirically used by the people of Southeast Sulawesi so that it can be developed into a standardized herbal preparation. The novelty of this study was exploring *C. alata* from Southeast Sulawesi, starting by determining antifungal activity against Malassezia furfur, identifying the main compounds, and then predict the effects of identified compounds on the lanosterol 14-alpha demethylase enzyme from *M. furfur*. Thus, the potential of *C. alata* against *M. furfur* was evaluated in vitro and in silico.

## 2. Results

### 2.1. Preparation of C. alata Leaf Extract and Fractions

The concentrated extract was dark green with a distinctive *C. alata* odor. The results of phytochemical screening showed that the leaf extract contained steroids, terpenoids, alkaloids, saponins, and flavonoids.

### 2.2. In Vitro Antifungal Activity

The results of antifungal activity test on *M. furfur* from extract and fractions of ethyl acetate and n-hexane can be seen in Table 1 and Table 2, respectively. The results showed that the extract had better antifungal activity compared to the fractions.

### 2.3. Identification of Major Compounds in Fractions

The LC-MS/MS technique was used to identify the ethyl acetate and n-hexane fraction compounds. Identification was based on molecular formulae and retention time (in minutes). Our results showed five major compounds for each fraction, but only three compounds were identified based on our database. The results of the tentatively identified compounds (TICs) using LC-MS/MS for the ethyl acetate fraction are shown in Figure 2 and Table 3, while the n-hexane fraction is shown in Figure 3 and Table 4.

### 2.4. In Silico Study

The three compounds listed in Figure 4 are compounds predicted to be present in the n-hexane fraction obtained from fractionation of *C. alata* extract. These three compounds are used as ligands to docked in the lanosterol 14-alpha demethylase enzyme. The results of molecular docking can be seen in Table 5. Visualization of the interaction between the enzyme and ketoconazole or with the three compounds can be seen in Figure 5 and Figure 6, respectively.

## 3. Discussion

The characteristic of plants is the production and storage of a complex mixture of secondary metabolites. The constituents of this mixture often belong to several secondary metabolite groups: for example, terpenoids accompanied by phenolics. We usually find small amounts of major secondary metabolites and several minor components, which are often biosynthetically related to the primary constituents [31,32]. Plants produce a variety of secondary metabolites that function as defense compounds against herbivores, plants, and microbes but also as signaling compounds [33]. Secondary metabolites contribute to the medicinal activity demonstrated by plants and indicate the involvement of natural products in the development of new medicines [34]. Thus, it is important that herbal plants provide scientific facts regarding the bioactive compounds and pharmacological tests of these plants [35]. The results of the phytochemical screening of the *C. alata* leaf extract were the same as the results of a study by Akinmoladun et al. [14]. Next, fractionation was carried out to separate secondary metabolites based on polarity [36].

Ethanol was chosen as the extraction solvent (menstruum) because, based on a study of the literature, ethanol extract activity is higher than water extract. This is caused by several factors: (1) Ethanol is more efficient at degrading nonpolar cell walls and causes polyphenols to be released from cells. (2) The polyphenol oxidase enzyme in water extract will degrade the polyphenols in the water extract. Meanwhile, this enzyme is not active in ethanol extract. (3) Water is a better medium for the development of microorganisms than ethanol [37]. (4) Ethanol more easily penetrates cell walls and extracts intracellular compounds in plants [38]. This study did not use methanol because although methanol’s extraction ability is better than ethanol, methanol is cytotoxic and can cause incorrect results [39].

The antifungal activity of the extract at the concentration of 50 to 400 mg/mL was higher than ketoconazole, 20 mg/mL, as the positive control (Table 1). The increase in the inhibition zone was proportional to the increase in extract concentration. The ketoconazole concentration was in accordance with the concentration of preparations circulating on the market, namely 2% [40]. This study did not compare the extracts and ketoconazole at the same concentration because, based on a study by Sulistyo et al. [41], at the same concentration as ketoconazole, namely 2%, the extract did not show antifungal activity against *M. furfur*. Thus, this study used the smallest extract concentration (50 mg/mL), in accordance with a study by Triana et al. [42], which formed an inhibition zone of 20.30 mm. The results of this study provided a bigger inhibition zone (29.89 ± 1.21 mm) than Triana et al.’s study. This was because we used DMSO as the solvent so that the extract had better solubility, then gave better diffusion and inhibited the growth of *M. furfur*. In addition to that, DMSO can dissolve polar and nonpolar compounds and immediately dissolves with water and cell culture media. DMSO also has a high boiling point, thereby increasing the accuracy of compound concentrations due to evaporation at room temperature [43]. This inhibitory capability aligns with previous studies indicating the antifungal potential of *C. alata* leaf extract against *T. verrucosum*, *E. floccosum*, and other microorganisms [12]. The antifungal activity of *C. alata* extract against *M. furfur* can be seen in Table 1. The results show that the antifungal activity is significantly different compared to DMSO as a negative control.

In this study, the antifungal activity of the n-hexane and ethyl acetate fractions were tested. The distilled water fraction was not tested for antifungal activity because *M. furfur* is a lipid-dependent fungus [22]. Meanwhile, the secondary metabolite content in the distilled water fraction was polar compounds because the nonpolar compounds were fractionated into the n-hexane and ethyl acetate fractions. The antifungal activity of *C. alata* fractions against *M. furfur* can be seen in Table 2. The increase in antifungal activity was proportional to the increase in fraction concentration (250 to 400 mg/mL). The n-hexane fraction had better antifungal activity compared to the ethyl acetate fraction (Table 2) because *M. furfur* is a lipid-dependent fungus [22]. Thus, nonpolar secondary metabolites in the n-hexane fraction diffuse more easily through the cell walls of *M. furfur* compared to secondary metabolites in the ethyl acetate fraction. Nonpolar compounds are more soluble in n-hexane, with a polarity index of 0.1, than in ethyl acetate, with a polarity index of 4.4 [44]. The results showed that the antifungal activity was significantly different compared to DMSO as a negative control. At the same concentrations (300 and 400 mg/mL), the inhibitory zones of the n-hexane and ethyl acetate fractions were smaller than the extract. This phenomenon may arise due to the relatively lower content of secondary metabolites in the n-hexane fraction compared to the ethanol extract [45]. Secondary metabolites interact with primary targets in cells, such as proteins, biomembranes, or nucleic acids. Meanwhile, alkaloids generally work on neurotransmitter receptors. Phenolic compounds and terpenoids react less specifically and attack many proteins by establishing hydrogen, hydrophobic, and ionic bonds, thereby modulating their 3D structure and consequently their bioactivity. The multitarget activity of many secondary metabolites explains the medical application of complex extracts of medicinal plants for more health disorders involving multiple targets [33].

It is estimated that only 20 to 30% of the 350,000 known plant species have been examined phytochemically in detail; therefore, the actual number of secondary metabolites in the plant kingdom likely exceeds 200,000 compounds [32]. The TIC results of the ethyl acetate fraction showed similar results to Rahman et al. [16], Liu et al. [18], and Promgool et al. [46], namely in flavonoids. Flavonoids inhibit fungal growth by the five following mechanisms: (1) Induced plasma membrane disruption. Ergosterol, as an important component of cell membranes, is a target for antifungal drugs, thereby disrupting cell membrane integrity and causing leakage of intracellular components [47,48] due to changes in membrane permeability [49]. Furthermore, overproduction of reactive oxygen species (ROS) causes severe oxidative stress in cells, causing progressive membrane permeabilization or damage to nucleic acids and oxidation of fatty acids and amino acids [50,51]. (2) Inhibition of cell wall formation. Fungal cell walls consist of β-glucan and chitin, so the synthesis of these compounds is an antifungal target [48,52]. The antifungal process is achieved based on cell wall deformation, which includes a decrease in cell size and an increase in membrane permeability [53]. (3) Induced mitochondrial dysfunction. Inhibition of the mitochondrial electron transport chain (ETC) reduces membrane potential. This inhibition occurs through the inhibition of the proton pump, which reduces ATP synthesis, and so the cell dies [52]. (4) Inhibition of cell division. Inhibition of cell division leads to the inhibition of microtubule polymerization, which inhibits mitotic spindle formation [52]. (5) Inhibition of the efflux pump. Efflux pumps are transporters in most living cells, playing a role in removing toxic substances from the body. This transporter can detoxify fungal cells through the removal of accumulated drugs. High expression of efflux pumps can lead to drug resistance. Therefore, inhibiting efflux pumps is an important goal for reducing drug resistance [54]. In this study, 5,7,2′,5′-tetrahydroxy-flavone was identified; there was a difference in the hydroxy positions from the results of a study by Rahman et al. [16], namely 3,5,7,4′-tetrahydroxy flavone and 2,5,7,4′-tetrahydroxy isoflavone. Meanwhile, kaempferol was identified in all three studies, while quercetin was only detected in the study by Liu et al. [18].

Quercetin (flavonol) has been reported to have antifungal activity and work synergistically with fluconazole, which is an inhibitor of fatty acid synthetase [55]. The antifungal activity of the kaempferol isolate is lower compared to the kaempferol-containing extract [56]. *C. alata* extract inhibits *Candida albicans*, *M. canis*, and *T. mentagrophytes* better than ketoconazole 200 mg [57]. Tetrahydroxy-flavone stimulates an increase in the permeability and physical alarm of the plasma membrane so that small molecules can diffuse; this causes the membrane to malfunction and depolarization, potassium ion leakage, and reduction of membrane fluidity, even triggering cell death [58,59].

The TIC results of the n-hexane fraction showed results similar to those of Saha et al. [60], namely for fatty acids. Various compounds identified in the LC-MS/MS analysis, such as 9-ene-methyl palmitate and stearidonic acid, have also been documented in previous studies as constituents of *C. alata* leaves [3]. Trichosanic acid, also known as punicic acid, is a long-chain omega-5 polyunsaturated fatty acid and a conjugated isomer of α-linolenic acid (CLnA) that exhibits structural similarities to conjugated linoleic acid (CLA) and α-linolenic acid (LnA). Previous investigations concerning the content of *C. alata* leaves did not report the presence of trichosanic acid. Notably, 9-ene-methyl palmitate, stearidonic acid, and trichosanic acid have also been identified in pomegranate seeds, with trichosanic acid as the primary fatty acid found in these seeds [61].

Fatty acids are organic acids that are characterized by having a carboxyl group (–COOH) and a methyl group (–CH_3_). The advantages of fatty acids as antifungals are natural and less likely to increase resistance in fungi [62]. The main mechanism of action of fatty acids as antifungals is that fatty acids insert themselves into the double layer of fungal lipid membranes, thereby disrupting membrane integrity, resulting in uncontrolled release of electrolytes and intracellular proteins, thus causing disintegration of the fungal cell cytoplasm. The chemical composition and pH of the environment are critical to the antifungal capabilities of these compounds [63]. Thibane et al. stated that some polyunsaturated fatty acids (PUFAs) increase the unsaturation ratio of cell membranes, which causes the accumulation of intracellular reactive oxygen species (ROS) and results in loss of mitochondrial membrane potential [64]. There are other mechanisms of fatty acid antifungal activity. For example, fatty acids inhibit topoisomerase I [65,66], an enzyme involved in DNA strand breaking and repair and topological changes necessary for cellular processes such as replication, transcription, and recombination [67]. Cis monounsaturated fatty acids are the most efficient in inhibiting topoisomerase I; with geometry, the position of the double bond and the length of the fatty acid carbon chain influence this inhibitory process [64]. Other mechanisms include the inhibition of fatty acid biosynthesis, such as 2-hexadecinic acid inhibiting triacylglycerol synthesis [68]. Another way fatty acids work is through inhibition of the N-myristoyltransferase (NMT) enzyme. NMT catalyzes the myristoylation reaction, which involves myristic acid. These reactions play a role in membrane targeting, protein–protein interactions, and signal transduction pathways. The incorporation of myristic acid analogues into fungal cells competes with myristic acid binding to NMT and disrupts the myristoylation reaction. Thus, it causes disruption of protein function, thereby inhibiting fungal growth [69].

Palmitic acid increases cellular toxicity, and even the rate of cell death increases in mutants of ole1, namely, the gene responsible for fatty acid desaturase through the induction of reactive oxygen species [70]. Fatty acid methyl esters, including methyl palmitate, have antifungal activity [71]. The antifungal mechanism of methyl palmitate is to damage cell walls and membranes [72]. Stearidonic acid (18:4 n-3) is an inhibitor of mitochondrial metabolism and biofilm formation in *Candida albicans* and *C. dubliniensis* [73]. There is no literature regarding the antifungal mechanism of trichosanic acid, which is a conjugated triene fatty acid. Still, the antifungal mechanism is thought to be the same as in other fatty acids.

An in silico study was conducted as an exploratory effort to identify compounds with optimal activity in inhibiting the lanosterol 14-alpha demethylase enzyme. Ketoconazole inhibits the lanosterol 14-alpha demethylase enzyme, which converts lanosterol into ergosterol, a component of fungal cell membranes [74]. The compounds conducted in the molecular docking process were selected based on that unveiled in the LC-MS/MS analysis of the n-hexane fraction (Figure 3). This was chosen because there has been no study regarding the interaction of fatty acid derivatives with the lanosterol 14-alpha demethylase enzyme. The results for bond energy and interactions with amino acids are shown in Table 3. While, the result of prediction of binding energy activity and amino acid interactions toward the lanosterol 14-alpha demethylase enzyme showed in Table 5.

Molecular docking investigations of the *M. furfur* enzyme have unveiled that the fatty acid derivatives engaged in interactions similar to ketoconazole (Figure 5). The potential for activity as a lanosterol 14-alpha demethylase inhibitor escalates as the binding energy becomes increasingly negative. Stearidonic acid and trichosanic acid exhibited the lowest affinity value, at −7.2 kcal/mol, followed by 9-ene-methyl palmitate. Visualization of the protein and compound interactions with the highest scores are depicted through 2D visualization in Figure 6.

9-Ene-methyl palmitate has hydrogen bonds with LEU46, while the stearidonic acid has fostered hydrogen bonds with SER456. Similarly, trichosanic acid has engaged in hydrogen bonds with SER329 (Table 5). Beyond hydrogen bonds, the ligands have also established Pi-Sigma and Pi-Alkyl interactions with other residues. Ligands exhibiting an increased count of hydrogen bonds and interacting residues demonstrate heightened binding affinity, indicative of effective protein inhibition.

The compounds identified through the LC-MS/MS results align with the findings from the in vitro and in silico assessments, signifying their antifungal activity. Previous studies have highlighted the potent bioactivity of these three fatty acid compounds in inhibiting various test microbes [61]. Ketoconazole impedes ergosterol biosynthesis by targeting the lanosterol 14-alpha demethylase within the lanosterol pathway. Ergosterol, a sterol present in fungi, governs membrane attributes like fluidity, rigidity, enzyme function, and asymmetry [75]. Moreover, ketoconazole stands out as one of the most efficacious drugs against *M. furfur*, with established inhibitory activity in diffusion tests and market availability [76].

## 4. Materials and Methods

### 4.1. Materials

*C. alata* was obtained from Kendari, Southeast Sulawesi, and identified in the Plant Taxonomy Laboratory, Faculty of Mathematics and Natural Sciences, Universitas Padjadjaran, with No. 276/HB/08/2021. *M. furfur* ATCC 14521 was obtained from the University of Indonesia. The ethanol, n-hexane, ethyl acetate, DMSO, and ketoconazole used were analytical-grade and purchased from Sigma Aldrich (Darmstadt, Germany). Potato dextrose agar (PDA) was bacteriology-grade and purchased from Oxoid (Hampshire, UK).

### 4.2. Preparation of C. alata Leaf Extract and Fractions

*C. alata* leaves were air-dried away from direct sunlight and subsequently ground into a coarse powder. A total of 472 g of dried leaves were macerated with ethanol for 24 h, then filtered. This maceration procedure was repeated twice. The collected extracts were concentrated using a rotary evaporator at 45 °C. This produced 43.2 g of thick extract with a yield of 9.15%. The phytochemical screening was carried out on extracts based on the method described by Tiwari et al. [39] and Benna et al. [77]. Subsequently, 20 g of concentrated extract was dissolved in distilled water at 50 °C, then fractionated through liquid–liquid extraction with n-hexane, followed by ethyl acetate [36]. The diluted fractions were concentrated and produced 7.65, 38.71, and 53.42% of the n-hexane, ethyl acetate, and distilled water fractions, respectively.

Identification of steroids. Liebermann Burchard’s test: The extract is evaporated until dry, then extracted with chloroform. Add 5 drops of acetic anhydride followed by sulfuric acid from the side of the test tube. Steroid presence is indicated by a purple to blue ring at the junction of the two fluids [77].

Identification of triterpenoids. Salkowski test: A total of 1 mL of the extract solution is added to 2 mL of chloroform, shaken, and filtered. Add 5 drops of concentrated sulfuric acid to the filtrate, shake, and let stand. Triterpenoid presence is indicated by a golden yellow precipitate [77].

Identification of alkaloids. The extract is dissolved in dilute hydrochloric acid and filtered. (a) Mayer’s test: The filtrate is reacted with Mayer’s reagent (potassium mercuric iodide). Alkaloid presence is indicated by a yellow precipitate. (b) Wagner’s test: The filtrate is made to react with Wagner’s reagent (iodine in potassium iodide). Alkaloid presence is indicated by brown or reddish sediment. (c) Dragendroff’s test: The filtrate is reacted with Dragendroff’s reagent (potassium bismuth iodide solution). Alkaloid presence is indicated by a red precipitate [39,77].

Identification of saponins. (a) Froth test: The extract is diluted with distilled water to 20 mL and shaken in a measuring cup for 15 min. Saponin presence is indicated by a foam layer, 1 cm thick. (b) Foam test: A total of 0.5 g of extract is shaken with 2 mL of water. Saponin presence is indicated by foam lasting for 10 min [39,77].

Identification of flavonoids. (a) Alkaline reagent test: The extract is made to react with 5 drops of sodium hydroxide solution. Flavonoid presence is indicated by a dark yellow color, which becomes colorless when dilute acid is added. (b) Lead acetate test: The extract is made to react with 5 drops of lead acetate solution. Flavonoid presence is indicated by a yellow precipitate [39,77].

Identification of phenols. Ferric chloride test: The extract is treated with 5 drops of ferric chloride solution. Phenol presence is indicated by a bluish black color [39,77].

Identification of tannins. Gelatin test: The extract is added to a 1% gelatin solution containing sodium chloride. Tannin presence is indicated by a white precipitate [39,77].

### 4.3. In Vitro Antifungal Activity Test

The disc diffusion method was employed to assess the antifungal efficacy of the extract and fractions. A mixture of 15 mL of PDA medium and 1 mL of fungal suspension was poured into petri dishes and allowed to solidify. Paper discs (6 mm in diameter), each loaded with a specific concentration of the extract, a positive control (ketoconazole), and a negative control (DMSO), were affixed onto the solidified PDA medium. The petri dishes were incubated for 72 h at 27 ± 2 °C. After the incubation period, each zone diameter was measured using a caliper, and the inhibition zone was recorded [78]. The inhibition zones were categorized into groups based on their strength, including weak, medium, strong, and very strong, as detailed by Martini and Eloff [79].

### 4.4. Identification of Major Compounds in the Fractions

The column used was the Intensity Solo 2 RP-18, 2.0 μm, 100 × 2.1 mm column (Bruker, Bremen, Germany). In the LC-MS/MS preparation, the sample was dissolved in methanol and the vial was put into the autosampler. For elution, a concentration of 5 μL was injected and a 0.250 mL/min flow rate was adjusted. Columns were stored at 35 °C and 10 °C for the sample. The mass spectrometer operated in negative mode with the following parameters: collision cell energy, 5.0 eV; nebulizer gas (N_2_) dry heater, 200 °C, −3.5 kV of ion spray voltage; dry gas (N_2_), 8.0 L.min^−1^; 2.0 bars and plate offset, 500 V. The internal calibration method was carried out with a solution consisting of 0.5 mL of 1N NaOH solution, 250 mL of H_2_O, 250 μL of formic acid, 750 μL of acetic acid, and 250 mL of iPrOH in the HPC mode. The data obtained from the LC-MS analysis were processed using Data Analysis software 4.4™ (Bruker, Bremen, Germany) to extract the mass spectral features of the raw data samples. Auto MS/MS mode was used to confirm fragment ions [80].

### 4.5. Molecular Docking

#### 4.5.1. Preparation of Enzyme

Preparation of the 3D enzyme structure involved the utilization of lanosterol 14-alpha demethylase, an enzyme pivotal in the synthesis of fungal cell walls. The protein sequence of Malassezia’s lanosterol 14-alpha demethylase in FASTA format was acquired from UniProt, identified by the code A0A3Q9XYP8 [81]. The enzyme’s preparation was conducted through homology modeling facilitated by the Swiss-Model^®^ web server (https://swissmodel.expasy.org/) [82]. Initiating the protein model creation process via homology modeling requires a template search conducted through the NCBI^®^ BLAST web server (https://blast.ncbi.nlm.nih.gov/Blast.cgi) to locate a pre-existing template akin to the protein [83,84]. Following this, the target sequence is aligned with the template and a model is generated by employing Swiss-Model^®^ (https://swissmodel.expasy.org/). The resultant model proteins are subsequently subjected to evaluation via Procheck^®^ (https://saves.mbi.ucla.edu/), and their structural integrity is determined through Ramachandran plots [85].

Subsequently, an active search was conducted to identify a suitable site based on cavity area using BIOVIA Discovery Studio Application 2017 to derive the X, Y, and Z coordinates. Following this, enzyme preparation was executed using BIOVIA Discovery Studio Application 2017, involving the removal of solvent residues (water), natural ligands, and other nonstandard residues from the enzyme. This process facilitated the isolation of natural ligand and enzyme files devoid of solvent and nonstandard residues, and these files were saved with the extension .pdb. Post-preparation, the enzyme structure was imported into the AutoDock Tools 1.5.6 software in the.pdbqt format [80].

#### 4.5.2. Preparation of Compounds

*C. alata* metabolites were utilized for the LCMS/MS analysis (Figure 4). The compound’s 3D structure was acquired from the website https://pubchem.ncbi.nlm.nih.gov/ (accessed on 24 October 2023) in .sdf format. The ligands’ format was then converted to .pdb using Open Babel [86]. Afterward, each ligand underwent optimization through AutoDock Tools, involving the specification of active torsions, after which it was saved in the ligand.pdbqt file format.

#### 4.5.3. Active Site Prediction

BIOVIA Discovery Studio software 2017 was employed to predict the active site and subsequently determine the site within the enzyme cavity. This active site’s attributes were then accessed to retrieve the coordinates for the grid box centers (x, y, and z).

#### 4.5.4. Docking Simulation

A folder named “config.txt” was prepared using Notepad, comprising information on enzymes, original ligands, and grid box center coordinates (x, y, and z). A new folder was then generated that contained enzymes and ligands in .pdbqt format. Furthermore, the compound molecules underwent docking using AutoDock Vina through the command prompt. Upon completion of the docking process, two new folders emerged, named “log.txt” and “out.pdbqt”. To glean insights into bond energies and inhibition constants, this output file was examined using the AutoDock Tools software [38].

#### 4.5.5. Interaction Analysis

Data analysis was conducted by assessing the bond energy derived from the molecular docking process. The binding energy value served as an indicator of the strength of the interaction between the enzyme and the ligand. A lower bond energy value signified a stronger bond between the ligand compound and the enzyme. The outcomes of the molecular docking were visualized using BIOVIA Discovery Studio to observe the formed interactions.

## 5. Conclusions

The extract, ethyl acetate, and n-hexane fractions derived from *C. alata* leaves exhibited antifungal activity against *Malassezia furfur*. The 9-Ene-methyl palmitate, stearidonic acid, and trichosanic acid within the n-hexane fraction demonstrated favorable affinity values toward the lanosterol 14-alpha demethylase enzyme based on docking results.

## Figures and Tables

**Figure 1 pharmaceuticals-17-00380-f001:**
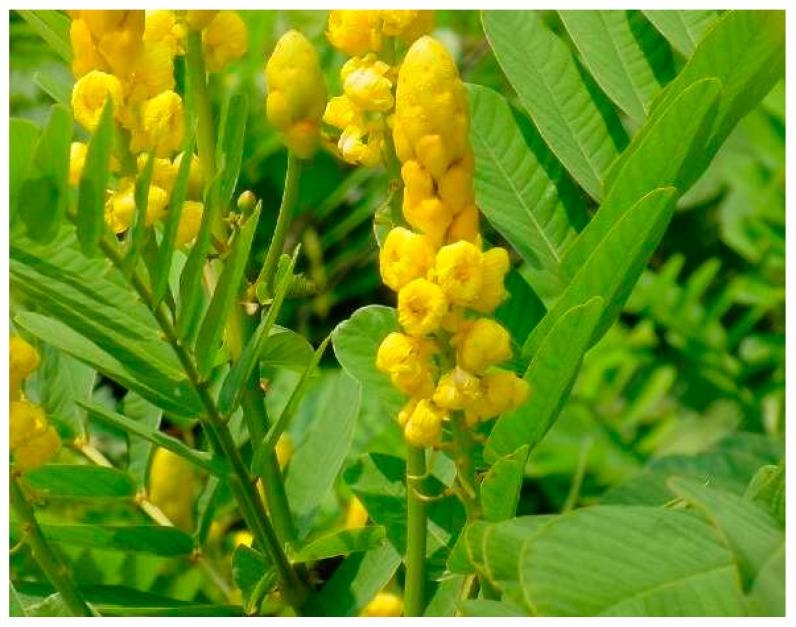
The plant of *C. alata*, also known as *Senna alata*, adapted with permission from Dave’s Garden [7]. Copyright 2021, copyright Dave’s Garden.

**Figure 2 pharmaceuticals-17-00380-f002:**
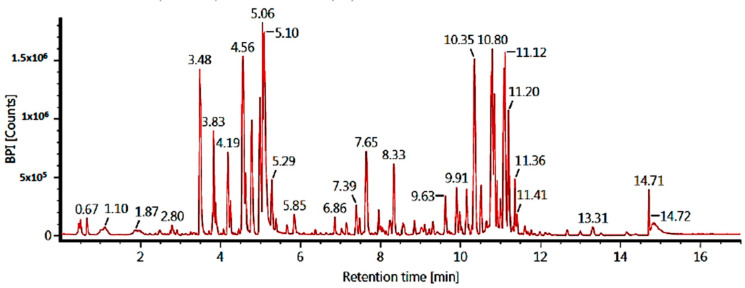
TICs of the ethyl acetate fraction of *C. alata* leaves (LC-MS/MS).

**Figure 3 pharmaceuticals-17-00380-f003:**
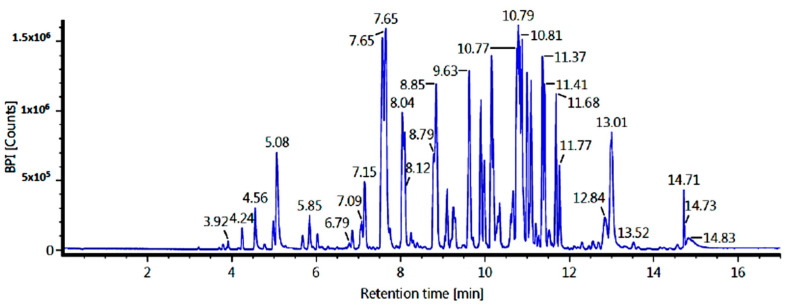
TICs of the n-hexane fraction of *C. alata* leaves (LC-MS/MS).

**Figure 4 pharmaceuticals-17-00380-f004:**
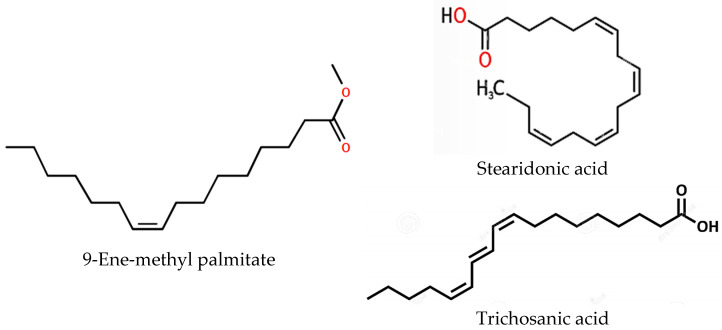
Structural representation of compounds in the n-hexane fraction of *C. alata* leaves from LC-MS/MS analysis.

**Figure 5 pharmaceuticals-17-00380-f005:**
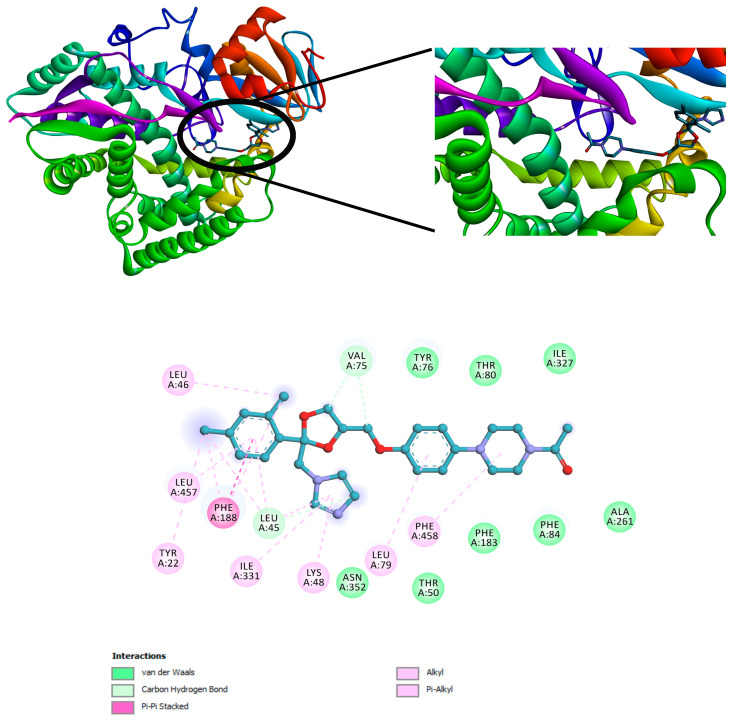
Interaction of ketoconazole with the lanosterol 14-alpha demethylase enzyme.

**Figure 6 pharmaceuticals-17-00380-f006:**
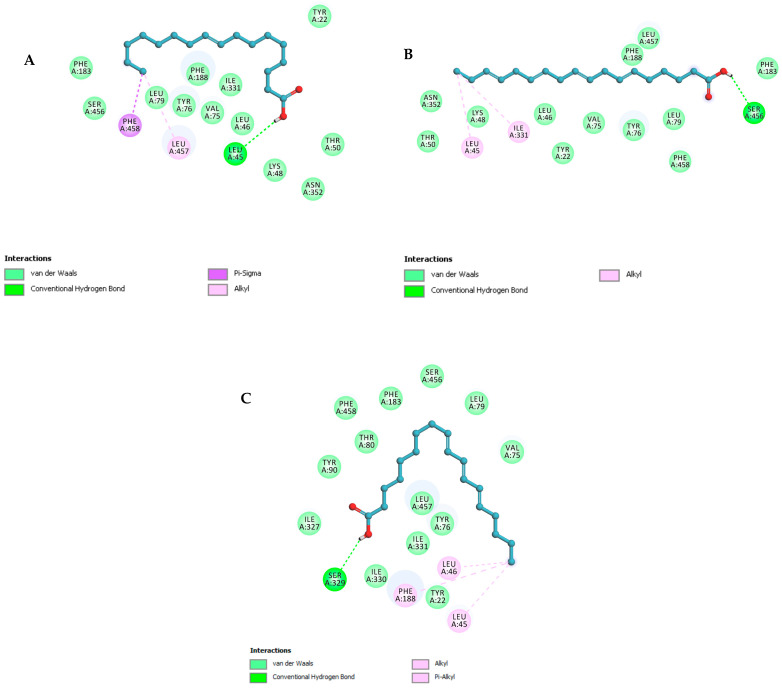
The 2D structural visualization of (**A**) 9-ene-methyl palmitate, (**B**) stearidonic acid, and (**C**) trichosanic acid.

**Table 1 pharmaceuticals-17-00380-t001:** Inhibition zone of *C. alata* leaf extract against *M. furfur*.

Compound	Concentration (mg/mL)	Inhibition Zone (mm)
Extract	50	29.89 ± 1.21 *
	100	30.33 ± 1.92 *
	150	32.11 ± 0.91 *
	200	32.67 ± 0.61 *
	300	33.78 ± 0.36 *
	400	39.11 ± 0.34 *
Ketoconazole	20	19.44 ± 1.27 *
DMSO	10	0

* Significantly different from DMSO.

**Table 2 pharmaceuticals-17-00380-t002:** Inhibition zones of n-hexane and ethyl acetate fractions from *C. alata* leaf against *M. furfur*.

Compound	Concentration (mg/mL)	Inhibition Zone (mm)
n-Hexane fraction	250	6.54 ± 0.41 *
	300	7.89 ± 0.17 *
	350	8.14 ± 0.15 *
	400	15.76 ± 0.09 *
Ethyl acetate fraction	250	4.99 ± 0.32 *
	300	9.04 ± 0.02 *
	350	9.32 ± 0.01 *
	400	9.68 ± 0.01 *
Ketoconazole	20	19.34 ± 1.27 *
DMSO	10	0

* Significantly different from DMSO.

**Table 3 pharmaceuticals-17-00380-t003:** Major compounds in the ethyl acetate fraction of *C. alata*.

Compound Name	Formula	Observed m/z	Neutral Mass (Da)	Observed RT (min)
5,7,2′,5′-Tetrahydroxy-flavone	C_15_H_10_O_6_	287.0541	286.04774	5.09
Kaempferol-3,7-diglucoside	C_27_H_30_O_16_	611.1624	610.15338	3.49
Quercetin	C_15_H_10_O_7_	303.0494	302.04265	4.59
Candidate mass C_35_H_36_N_4_O_5_	C_35_H_36_N_4_O_5_	593.2758	592.26857	10.35
Candidate mass C_36_H_38_N_4_O_7_	C_36_H_38_N_4_O_7_	639.2825	638.27405	10.78

**Table 4 pharmaceuticals-17-00380-t004:** Compounds in the n-hexane fraction of *C. alata* leaves.

Compound Name	Formula	Observed m/z	Neutral Mass (Da)	Observed RT (min)
9-Ene-methyl palmitate	C_17_H_32_O_2_	291.2306	268.24023	8.84
Stearidonic acid	C_18_H_28_O_2_	277.2154	276.20893	7.65
Trichosanic acid	C_18_H_30_O_2_	279.2311	278.22458	8.07
Candidate mass C_36_H_38_N_4_O_7_	C_36_H_38_N_4_O_7_	639.2819	638.27405	10.77
Candidate mass C_34_H_34_N_4_O_5_	C_34_H_34_N_4_O_5_	579.2615	578.25292	11.37

**Table 5 pharmaceuticals-17-00380-t005:** Prediction of binding energy activity and amino acid interactions toward the lanosterol 14-alpha demethylase enzyme.

Compound	∆G (kcal/mol)	Amino Acid Residue Hydrogen Bond	Other Amino Acid Residues Involved
Ketoconazole (positive control)	−7.9	TYR90, HIS415, PRO409	ALA261, PRO326, ILE327, PHE410, CYS417
9-Ene-methyl palmitate	−6.5	LEU46	PHE458, LEU457
Stearidonic acid	−7.2	SER456	ILE331, LEU45
Trichosanic acid	−7.2	SER329	ILE330, ILE327, ILE331, LEU457, TYR76, TYR22, TYR90, THR80, PHE458, PHE183, SER456, LEU79, VAL75

## Data Availability

Data are contained within the article.

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
