# Peer review of "Cassia alata L.: A Study of Antifungal Activity against Malassezia furfur, Identification of Major Compounds, and Molecular Docking to Lanosterol 14-Alpha Demethylase"

_pharmaceuticals, 2024, doi:10.3390/ph17030380_

Round 1

Reviewer 1 Report

Comments and Suggestions for Authors

This paper provides a well-written account of research on Cassia extract.  The background is described thoroughly, describing previous work on this species.  The extract preparation is described in adequate detail, although on line 84 the authors should specify that ethanol is used to make the extract, this is important as the term "ethanol extract" used later in the manuscript is unclear unless one reads the methods, something that should not be assumed.  The authors then describe crude separation into hexane and ethyl acetate fractions.  Biological assays of the extracts are then performed, showing significant activity.  Each extract is then analyzed by GC-MS, revealing a few matches with known compounds, although many peaks are not identified.

At this point, I would normally expect that each of these compounds would be assayed, to determine which (if any) were active; or else further fractionation would be undertaken.  I still recommend that the authors consider whether this is practical.  In the absence of such a correlation between structure and activity, the work is weak.  However, the authors do at least perform an in silico assay, the results of which suggest that the activity of the crude extract derives at least in part form the identified compounds.  This correlation, while not as strong as could be possible if authentic pure compounds had been assayed as suggested above, moves the manuscript from the "too preliminary" category into the acceptable range.  For this reason, I recommend acceptance, with the following minor points addressed:

Please fix the structures at line 113. Stearidonic acid does not have an acid group, trichosanic acid should have a H con the carboxyl group (or should be described as the anion), and 9-ene-methyl palmitate is not a proper nomenclature.

Fig., 5c: trichosanic acid has cis double bonds, yet the bound structure appears to be entirely trans.  Likewise, the stereochemistry of the unsaturated palmitate.

Line 149: the claim of higher activity for the extract than the ketoconazole is not reasonable, since although the zone of inhibition is larger, so is the concentration.

Line 154-156: I think there are many possible reasons for the difference described here, not just the suggested reason, e.g. concentrations may vary, co-operative effects, or that the active component is not one of the identified species.  I recommend removing this sentence, or expanding it.  

Line 284: "active side" presumably means site.

Comments on the Quality of English Language

Generally fine.

Author Response

Dear reviewer

Thank you for reviewing our manuscript

We have improved the manuscript according to the suggestions and comments provided

Hopefully the revisions meet the suggestions and comments

-best regards-

Reviewer 2 Report

Comments and Suggestions for Authors

The present manuscript is dedicated to the study of the anti-Malassezia furfur properties of leaf extract fraction from Cassia alata leaves, the major compounds in the fractions and molecular docking of these compounds to lanosterol 14-alpha demethylase. The rsults are of interest, however there are several concerns and questions, as follows:

1.      Page 1, line 39: antraquinones are not flavonoids.

2.      Page 2, lines 73 – 74: Several studies report also on the antifungal activity of the plant. Please cite some of these studies, e.g.

- Edegbo, E., Okolo, M. L. O., Adegoke, A. S., Omatola, C. A., Idache, B. M., Abraham, J. O., ... & Muhammed, D. (2023). Phytochemical screening and antifungal activity of Cassia alata (Linn.) crude leaf extracts. Afri J Microbiol Res, 17(8), 176-183.

- Phongpaichit, S., Pujenjob, N., Rukachaisirikul, V., & Ongsakul, M. (2004). Antifungal activity from leaf extracts of Cassia alata L., Cassia fistula L. and Cassia tora L. Songklanakarin J Sci Technol, 26(5), 741-748.

3.      Page 2, lines 84 – 89: These numerical data belong to the descriptiion of experiments in Materials and Methods. Here belong only the results of the phytochemical screening.

4.      The way identification was achieved should be described in Materials and Methods. Were reference compounds used? Which ones? Where did they come from?

5.      Page 6, line 148 – 149: How was this prediction made? Why was the total ethanol extract not tested as well?

6.      Page 6, line 222: Voucher specimen?

7.      Page 10, line 307: add “based on docking results”

Comments on the Quality of English Language

The present manuscript is dedicated to the study of the anti-Malassezia furfur properties of leaf extract fraction from Cassia alata leaves, the major compounds in the fractions and molecular docking of these compounds to lanosterol 14-alpha demethylase. The rsults are of interest, however there are several concerns and questions, as follows:

1.      Page 1, line 39: antraquinones are not flavonoids.

2.      Page 2, lines 73 – 74: Several studies report also on the antifungal activity of the plant. Please cite some of these studies, e.g.

- Edegbo, E., Okolo, M. L. O., Adegoke, A. S., Omatola, C. A., Idache, B. M., Abraham, J. O., ... & Muhammed, D. (2023). Phytochemical screening and antifungal activity of Cassia alata (Linn.) crude leaf extracts. Afri J Microbiol Res, 17(8), 176-183.

- Phongpaichit, S., Pujenjob, N., Rukachaisirikul, V., & Ongsakul, M. (2004). Antifungal activity from leaf extracts of Cassia alata L., Cassia fistula L. and Cassia tora L. Songklanakarin J Sci Technol, 26(5), 741-748.

3.      Page 2, lines 84 – 89: These numerical data belong to the descriptiion of experiments in Materials and Methods. Here belong only the results of the phytochemical screening.

4.      The way identification was achieved should be described in Materials and Methods. Were reference compounds used? Which ones? Where did they come from?

5.      Page 6, line 148 – 149: How was this prediction made? Why was the total ethanol extract not tested as well?

6.      Page 6, line 222: Voucher specimen?

7.      Page 10, line 307: add “based on docking results”

Author Response

(The authors gave the same response as above.)

Reviewer 3 Report

Comments and Suggestions for Authors

In this manuscript, Mekar and Co-workers describe the extraction of the leaves of Cassia alata. Three extractions were done (EtOAc, Hex, and water). Both EtOAc and Hexenes extracts were screened against antifungal activity (Malassezia furfur). It was observed that the original extract presented around 30 mm of inhibition zone (ZoI) at concentrations ranging from 50-400 mg/mL, while the control Ketoconazole showed 20 mm of ZoI at only 20 mg/mL.  Then, the extracts were evaluated, giving from 6.5-16 mm for the hexanes fraction and 5-10 mm for the ethyl acetate fraction. These values are lower than the standard ketoconazole 20 mm at an even lower concentration. Then, LC-MS/MS was used to identify the potential compounds responsible for the antifungal activity. From this study, six compounds matched known standards (e.g., quercetin, 9-ene-methyl palmitate, stearidonic acid, and trichosanic acid). Furthermore, molecular docking studies were performed for the standard ketoconazole and the best adducts  (i.e., 9-ene-methyl palmitate, stearidonic acid, and trichosanic acid). The manuscript is well-presented, but the novelty is low. There is a similar study (see ref 3). Also, the authors claim that "the results of the disc diffusion method showed that the C.alata leaf extract was better  inhibitory efficacy against M. furfur compared to ketoconazole as the positive control." (see lines 139-140). However, from the values reported in Tables 1 and 2 that is not the case. The values may look larger but they are at higher concentrations.  

Minor correction. 

1. figure 3. Two structures of the compounds are missing atoms.

2. Add a picture of the plant in the introduction section

3. Add a figure with the structures of the major known compounds of this plant.  

Comments on the Quality of English Language

The quality of English is fine, just minor editing is required. 

Author Response

(The authors gave the same response as above.)

Round 2

Reviewer 3 Report

Comments and Suggestions for Authors

The authors have addressed prior comments.